# Emergency Cervical Cerclage

**DOI:** 10.3390/jcm10061270

**Published:** 2021-03-18

**Authors:** Magdalena Wierzchowska-Opoka, Żaneta Kimber-Trojnar, Bożena Leszczyńska-Gorzelak

**Affiliations:** Chair and Department of Obstetrics and Perinatology, Medical University of Lublin, 20-090 Lublin, Poland; zkimber@poczta.onet.pl (Ż.K.-T.); b.leszczynska@umlub.pl (B.L.-G.)

**Keywords:** recurrent pregnancy loss, cervical insufficiency, premature birth, painless cervical dilatation, emergency cervical cerclage, chorioamnionitis, preterm premature rupture of membranes

## Abstract

Despite the progress of medicine in the last decades, recurrent pregnancy loss, premature birth, and related complications are still a vast problem. The reasons for recurrent pregnancy loss and preterm delivery are diverse and multifactorial. One of the main reasons for these complications is cervical insufficiency, which means that the cervix is weak and unable to remain closed until the date of delivery. It manifests as painless softening and shortening of the cervix without contractions. The aim of the study was to review the available literature on rescue sutures, which are an emergency treatment in pregnancies with premature cervical dilatation and protrusion of the fetal membranes in the second trimester of pregnancy. This review confirms that emergency cerclage reduces the rate of preterm birth in patients with advanced cervical insufficiency. This procedure prolongs gestational age and improves the chances of survival of the newborn without increasing the risk of chorioamnionitis and preterm premature rupture of membranes.

## 1. Introduction

Despite the progress of medicine in the last decades, recurrent pregnancy loss, premature birth, and related complications are still a vast problem [1,2,3]. Nevertheless, the development in neonatal care has led to a spectacular increase in neonatal survival. The reduction in neonatal mortality is mainly the result of the increased survival of very preterm babies. It is associated with regionalization of care for high-risk mothers and preterm infants, improved interdisciplinary cooperation of specialists, and development of neonatal intensive care. Despite this, premature babies are still at greater risk of cerebral palsy, impairment of mental development, visual and auditory disturbances, and neurological abnormalities associated with cognitive function and behavioral disorders [4,5,6,7,8,9,10,11,12].

The reasons for the recurrent pregnancy loss and preterm delivery are diverse and multifactorial. The most common diagnoses of recurrent pregnancy loss include Asherman syndrome (intrauterine adhesions), cervical insufficiency, and uterine fibroids, accounting for 47% of the patients [13]. The incompetent cervix is a well-recognized cause of mid-trimester miscarriage, recurrent pregnancy loss in the mid-trimester, and preterm labor presenting with bulging membranes in the absence of significant uterine contractility or rupture of membranes [14]. Cervical insufficiency affects 1% of all pregnancies and 8% of women with recurrent mid-trimester losses [15]. In a large Danish register-based cohort study Sneider et al. [16] found that the overall recurrence rate of second trimester miscarriage or extreme preterm delivery (16^+0^ to 27^+6^ gestational weeks) was 7%, but it differed significantly by phenotype. The highest rate, 28%, was found in cervical insufficiency [16].

The cervix plays a very important role in maintaining pregnancy. It is also a mechanical barrier that prevents exposure and prolapse of the fetal membranes and in combination with the mucus plug protects against ascending infection [17,18]. Cervical insufficiency means that the cervix is weak and unable to remain closed until the date of delivery, manifesting itself in painless softening and shortening of the cervix without contractions [18,19,20]. Cervical failure is related to its premature ripening. The processes of shortening and dilatation of the cervix are physiological processes during labor but when initiated before 37 weeks of pregnancy, miscarriage, or premature birth may occur [11,21,22]. Several factors are implicated in cervical insufficiency including: congenital changes in the internal structure of the cervix, acquired mechanical injuries, and inflammatory processes, leading to its early shortening and dilatation [23].

In recent years, an increase in the incidence of endocrine pathology, multiple pregnancy, connective tissue dysplasia, and pregnancies after in vitro fertilization has resulted in a higher number of the incidence of functional cervical insufficiency [24]. The pathophysiology of this condition may be associated with a constitutional deficiency of the components of the cervical connective tissue [25], as well as a natural biologic variation in cervical collagen, cervical elastin, and other structural components of cervical connective tissue that resist softening, effacement, and dilation. Congenital factors include: pathologies of collagen synthesis-disorders of type I collagen regulation, i.e., Ehlers–Danlos syndrome (which explains the large number of cases of cervical failure within families) and uterine anomalies [23,26]. In addition to congenital factors, cervical insufficiency is also influenced by the loss of connective tissue in the cervix following operations, e.g., after cervical conization, after mechanical expansion of the cervical canal during curettage of the uterine cavity or after an incorrectly repaired postpartum cervical laceration [23,26]. Emergency cesarean section at full dilatation is also implicated in subsequent cervical insufficiency. New theories suggest that inflammation and changes in the normal vaginal microbiome may contribute to changes in the structural integrity of the cervix [27]. Furthermore, Tanner et al. [28] demonstrated that racial and ethnic differences exist in the frequency of diagnosis of insufficient cervix. The odds of being diagnosed with cervical insufficiency among black women were almost three times that of white women, after controlling for a number of potential confounders. Interestingly, the rates of other risk factors found to be independently associated with this condition, i.e., prior pregnancy termination and prior cervical procedures, were also more common among black women [28].

The cervical cerclage strengthens the weak cervix, maintains its length and preserves the mucus plug at the cervical opening—protecting against ascending infection. Elective prophylactic cervical cerclage is usually performed in women who have a history of spontaneous second-trimester miscarriages and preterm delivery or those who underwent cervical procedures, such as conization, that may cause cervical incompetence. In the Danish study, prophylactic cerclage applied before 16 weeks of gestation was associated with a significant reduction in recurrence rate of second trimester miscarriage or extreme preterm delivery [16]. Vaginal cerclage was associated with a significant reduction (adjusted odds ratio (OR) 0.47; 95% CI 0.29–0.76) and transabdominal cerclage with an even greater reduction (adjusted OR 0.14; 95% CI 0.03–0.61) [16]. Transabdominal cervical cerclage may be inserted during the first trimester of pregnancy or preconceptually [29,30].

Cervical cerclage may also be performed when there is evidence of a short cervix or cervical shortening on ultrasound. The conclusions from many studies confirm that the cervical suture inserted in women with shortened cervix on ultrasound examination and a low risk of preterm labor in the history does not reduce the frequency of preterm labor [15,31,32]. Its effectiveness increases in women with a shortened cervix and with a burdened obstetric history [33,34,35], but it decreases with greater dilatation [36].

Less often, a rescue cervical suture may be inserted when the patient presents with a cervix that is already dilated with the membranes bulging into the vagina but no signs of labor, infection or heavy vaginal bleeding [37] (Figure 1). Cervical dilatation and protrusion of the fetal membranes in the second trimester of pregnancy put pregnant women at high risk of miscarriage or premature birth. A dilated cervix is identified on speculum examination and physical examination or transvaginal ultrasound. A rescue cerclage is an emergency treatment, which extend the duration of pregnancy [38].

The aim of the study was to review the available literature on emergency sutures.

## 2. Emergency Suture Procedure

The emergency cerclage represents the main treatment strategy in case of cervical insufficiency and protruding membranes in pregnant women [39]. However, specialists with high experience should decide on qualifying for its placement. This procedure should be carried out by a well-trained operator.

The emergency suture was defined as a suture placed over the cervix with a premature opening and possible bulging of the membranes into the vagina, after excluding a rupture of the membranes. Placement of this suture is an emergency procedure aimed at prolonging the duration of pregnancy [40]. Nevertheless, cervical emergency suture may also increase the risk of infection due to exposure of the membranes to vaginal bacteria, and its effectiveness and safety remain controversial.

Previous studies have found that microorganisms are present in the amniotic cavity in 8–52% of patients with cervical insufficiency [41,42,43,44,45,46,47,48]. Oh et al. [49] concluded that administration of antibiotics (ceftriaxone, clarithromycin, and metronidazole) in patients with cervical insufficiency and intra-amniotic inflammation or intra-amniotic infection can improve the treatment success.

On the other hand, Chatzakis et al. in their meta-regression analysis showed that there was no effect of antibiotics administration on the outcome of pregnancy prolongation [39]. Most studies reported that antibiotics administration was at the discretion of the managing clinicians [50,51,52,53,54]. Due to the lack of randomized control trials, these observations should be viewed with caution, as the quality of evidence is low to very low.

The procedure of insertion an emergency suture is a technically difficult procedure. The protruding membranes in the cervix make it difficult to place the cervical suture properly and carry the risk of iatrogenic rupture of the membranes during surgery. In the case of emergency sutures, there are many described ways to apply them, but no studies have shown advantage of any technique [55,56,57].

Cervical cerclage involves the positioning of a suture around the neck of the womb. The aim is to provide mechanical support to the cervix and keep the cervix closed during the pregnancy. A stitch, usually silk, tape, or other nonabsorbable material, is inserted around the cervix in three or four bites, enclosing it [15]. Placing a rescue suture requires a slightly different procedure than the standard procedure. In order to avoid preterm premature rupture of membranes, the fetal membranes should be moved above the planned suture site. Over the years, moist swabs have been used for this purpose, but a less invasive method has been developed. Filling the bladder with physiological saline in a patient positioned in the Trendelenburg position turned out to be technically effective for draining a prolapsed fetal membranes [58,59,60] (Figure 2). When the bladder is full, it lifts the inferior pole of the fetal membranes, causing the membranes to withdraw from the vagina. A cervical suture is placed as high as possible over the cervix according to the McDonald’s technique [15].

The precursors of this method were Scheerer et al. [61]. This method is more sparing than gently repelling herniated membranes with a damp cloth or gauze, and carries a lower risk of infection. The possibility of keeping the mucus plug in the cervical opening is indicated as a potential advantage of this procedure [61].

Modifications to this method are also made. Debby et al., to replace gently the protruding membranes into the uterus, used Foley catheter filled with 30 mL saline, which they removed after emptying the balloon at the end of the surgery immediately followed by tightening of the suture [62].

Son et al. proposed a special uniconcave balloon (Figure 3), but in their study, pregnancies after emergency sutures were prolonged up to 37 weeks only in 20.9% and in 25% it ended before 24 weeks [63]. This tool is convenient to use for the operator, but the results are disappointing [64].

Min Lv and co-authors reviewed in detail procedures of the balloon tamponade assisted emergency cerclage [65]. They assessed that rescue cerclage assisted by the balloon tamponade was favorable performed in 39 women with cervical dilatation and protruding fetal membranes. The medium diameter of the bulging membranes was 2 cm with maximum range up to 10 cm. Pregnancies were prolonged by 8 to 138 days with a medium time of prolongation 29 days and there were no perioperative complications as infection, injury, or bleeding. Authors concluded that rescue cerclage is an effective way to extend pregnancy in patients with advanced cervical dilatation and herniating membranes. They observed that the balloon tamponade is a useful device which effectively replaced fetal membranes into the uterine cavity to enable the placement of an emergency cerclage [65].

Some surgeons use amniocentesis to reduce the tension in the fetal membranes that invades the vagina. Performing amniocentesis with preoperative amnioreduction to decompress the fetal membranes does not seem to be a recommended method. Too little data were available to justify the routine use of this procedure. There are no randomized trials confirming the effectiveness of such a procedure. This method may additionally be a factor that increases the number of complications [57]. Although the obtained amniotic fluid can be used simultaneously for bacteriological examination, it has not been unequivocally demonstrated that such a procedure results in an improved prognosis [57].

In 2020, Medjedovic et al. [66] presented successful outcomes in one patient with 3 cm cervical dilatation and concomitant prolapse of the fetal membranes in whom 120 mL of clear amniotic fluid were removed prior to McDonald emergency cerclage. The authors concluded that favorable findings are possible after appropriate adjunctive therapy with antibiotics, bed regimen and regular gynecological supervision [66]. This procedure requires further research.

## 3. Rescue Suture Effectiveness

Due to the ethical reasons, the majority of data concerning the effectiveness of emergency cervical cerclage come from retrospective analyses. Bulging of the membranes into the vagina represents a huge obstetrical problem. Therefore, each retrospective study regarding this matter seems valuable.

The Spanish authors analyzed the results of 39 patients who underwent emergency cervical cerclage. The average prolongation of pregnancy in these patients was 49.1 days, with a mean delivery time of 28.6 weeks and neonatal survival of 82.4% [67]. Slightly less favorable results were obtained in the analysis of 23 patients described by Caruso et al. [68]. The average prolongation of pregnancy in these patients was 4 weeks, and the survival rate was 46%, with the mean time of delivery at 25 weeks of pregnancy and the average neonatal birth weight of 700 g. The authors nevertheless considered this a good result since cervical dilatation and protrusion of the fetal membranes represent an essential clinical problem [68].

Positive results of the emergency suture application were also confirmed in a study of Ciancimino et al. [69]. A total of 12 patients underwent emergency suture placement. Pregnancy was extended by an average of 89.9 days, and neonatal survival was 83.3% [69]. Equally good results were obtained by Cavus et al. [70], who analyzed pregnancy outcomes in 20 patients who experienced cervical cerclage in the second trimester with an average of 4.3 cm cervical dilation. The mean time between the procedure and delivery was 13.8 weeks. The total live birth rate was 90% [70].

Rescue suture has been recommended by Mitra et al. as well. They observed the average prolongation of pregnancy of 12 weeks in 40 patients. A total of 31 of 40 pregnancies were continued to 28 weeks of gestation and 23 were carried to 34 weeks of gestation or more. The authors achieved a newborn survival rate of 83% [71].

In order to verify the effectiveness of the procedure, the use of the emergency suture was compared with bed rest in several studies. The conservative management with bed rest were performed in patients who have refused the operative procedure. Many authors have observed significant benefits resulting from the rescue cervical cerclage in comparison to the expectant management [50,72,73,74,75,76,77]. These outcomes are presented in Table 1.

Canadian researchers analyzed the data of 12 patients undergoing emergency cervical cerclage. They obtained the average pregnancy prolongation of 7 weeks [78]. Additionally, the authors conducted a literature review on emergency sutures from 1995–2005 describing 25 studies with 638 women. The average duration of pregnancy prolongation in these patients was 7 weeks and 1 day. The mean neonatal survival rate was over 70%. The incidence of premature rupture of membranes was 29% of all pregnancies. The authors summarizing the collected data suggested that the emergency cerclage, under ideal conditions, can significantly extend the pregnancy [78].

A meta-analysis by Christos Chatzakis and co-authors was published in 2020. The authors reviewed 38 studies regarding emergency cervical cerclage and assumed 12 observational analysis with 1021 patients [39]. Emergency cerclage had more beneficial results than the expectant management before 28 and 32 weeks of pregnancy. The operative treatment was superior to expectant management for pregnancy prolongation (by an average of 47 days), older gestational age at delivery (with difference over 5 weeks), the risk of neonatal hospitalization in intensive care unit and fetal mortality rate. Unfortunately, these favorable results were associated with very low and low quality in statistical analyses. However, it should be emphasized that the risk of the chorioamnionitis and premature rupture of membranes during or after the surgical intervention is similar to those of the conservative management [39].

## 4. Risk Factors of the Emergency Suture Failure

Appropriate perioperative procedures may reduce the risk of spontaneous premature delivery. Unfortunately not all risk factors can be eliminated (Table 2), there are beyond the possibilities of the operators. It seems that the only modifiable factor is treatment of genital tract infections. Undoubtedly, the most successful situation is a lack of vaginal colonization with pathogenic microorganisms and the absence of signs of infection. Considering that operators are obligated to make a quick decision about qualifying patient to rescue cerclage taking into account the increasing dilatation, targeted treatment in the preoperative period is usually impossible.

Fuchs et al. used multivariate logistic regression methods to develop a score for assessing the risk of early preterm delivery before 32 weeks in women with singleton pregnancies receiving emergency cervical cerclage [64]. The score, ranging from 0 to 15 points, was based on the following four criteria independently associated with early preterm delivery: obstetric history; cervical dilatation; membranes bulging into the vagina; and infection (Table 3). Each score value was associated with a predicted probability of early preterm birth. The authors found that a history of second-trimester pregnancy loss, nulliparity, a cervix dilated more than 4 cm, membranes bulging into the vagina, and infection (i.e., white blood cells (WBC) ≥ 13,600/ mm^3^ or C-reactive protein (CRP) > 15 mg/L) are associated with emergency suture failure [64].

Ito et al. demonstrated that elevated serum inflammatory markers before the procedure are associated with its failure [79]. CRP value and WBC are recognized predictors of subclinical chorioamnionitis [80]. The authors observed that peripheral CRP levels (≥4 mg/L) and WBC counts (≥10,000/mm^3^) were associated with a significantly decreased likelihood of delivery at and after 28 weeks gestation [79].

Some authors expect that Gram’s method and culture of amniotic fluid are imperfect techniques to detect infection in patients with cervical insufficiency that are qualified to emergency cerclage procedure [81]. Proteomic profiling of amniotic fluid could be better solution in that cases. There are different expression of proteins between patients with delivery before one week from inserting cerclage and those which had delivery later. Patients with shorter prolongation of pregnancy had activated inflammatory response, chemotaxis of immune cells, and inhibited bacterial growth. These preliminary results indicate that the proteomic profiling of amniotic fluid may be an effective predictor of cervical insufficiency outcome [68]. More research on this method is required.

Lee et al. undertook the task to discover new amniotic fluid markers that can be important prognostic factors in patients with cervical incompetence [82]. They performed a retrospective cohort study on 40 patients with rescue cerclage who underwent amniocentesis and used label-free liquid chromatography-tandem mass spectrometry (LC-MS/MS) to recognize components of amniotic fluid. A total of six selected biomarkers were verified by enzyme-linked immunosorbent assays (ELISA). Researchers found that amniotic fluid of patients that had spontaneous preterm delivery before 34 weeks of gestation after cerclage placement presented greater levels of myeloperoxidase, lactoferrin, glucose-6-phosphate isomerase, lipocalin-2, and lymphocyte cytosolic protein 1. The authors associate their inherence with a poor prediction after rescue cerclage for cervical incompetence [82].

The most significant risk factors for failure of this procedure are advanced dilation and effacement of the cervix and bulging of the membranes into the vagina. In a retrospective study summarizing the results of inserting an emergency cerclage in 130 pregnant women, it was observed that complications as chorioamnionitis and failure of the therapy are related to the cervical dilatation over 5 cm and protruding membranes into vagina [83]. Similar conclusions were made in the study conducted by Uzun Cilingir et al., which included 21 pregnant women with bulging membranes and dilatation of cervix over 4 cm [36]. The authors found that operative treatment is not a reasonable procedure in case of such an advanced dilatation. It is associated with a high complication level and moderate prolongation of the pregnancy [36].

Multiple pregnancy is also a risk factor for method failure. Chun and co-authors reported lower rate of neonatal survival in twin pregnancies treated with emergency cerclage than in singleton pregnancies. In twin pregnancies with cervical incompetence, this procedure might be taken into consideration only as a rescue treatment [84]. Although the randomized controlled trials are unavailable, Cilingir et al. suggest that this procedure in patients with twin pregnancies and cervical shortening less than 15 mm should be performed [85]. The emergency cervical cerclage can be an alternative of treatment only for selected women with twin pregnancies with advanced cervical dilatation and exposed fetal membranes.

Natalie Suff and co-authors conducted a retrospective observational study of 35 patients in single pregnancy with protruding membranes between 18 and 23^+6^ weeks of pregnancy [86]. Authors assessed predictive level of fetal fibronectin concentration in vaginal mucus for preterm delivery risk in those women. The measurement of quantitatively fetal fibronectin was performed day before the surgery. The mean gestational age at delivery was 29^+3^ weeks. The pregnancies were prolonged after the procedure by a mean of 65.5 days. In total, 12 patients had delivery within 4 weeks from procedure of rescue cerclage placement. 60 percent of those women had a level of fetal fibronectin over 500 ng/mL, and all had a delivery before 37 weeks of pregnancy. Patients with fetal fibronectin levels below 10 ng/mL did not have delivery within 4 weeks of cerclage insertion, most of them delivered at term. The authors observed a significant difference in the level of fetal fibronectin between patients with preterm delivery and patients with delivery on time. They concluded that quantitative fetal fibronectin is an important predictor of premature birth in case of pregnancies with protruding membranes treated previously with rescue cervical cerclage [86]. It seems that this method can be valuable in qualifying patients for rescue cerclage procedure.

## 5. Conclusions

Emergency cerclage reduces the rate of preterm birth in patients with painless cervical dilatation and protrusion of the fetal membranes. This procedure prolongs gestational age and improves the survival of the newborns. However, it does not increase the risk of chorioamnionitis and preterm premature rupture of membranes. Due to the limited number of randomized control trials and low quality of evidence, in spite of the extremely favorable estimates for cerclage, these results should be viewed with caution. They should therefore be confirmed by more extensive clinical trials.

## Figures and Tables

**Figure 1 jcm-10-01270-f001:**
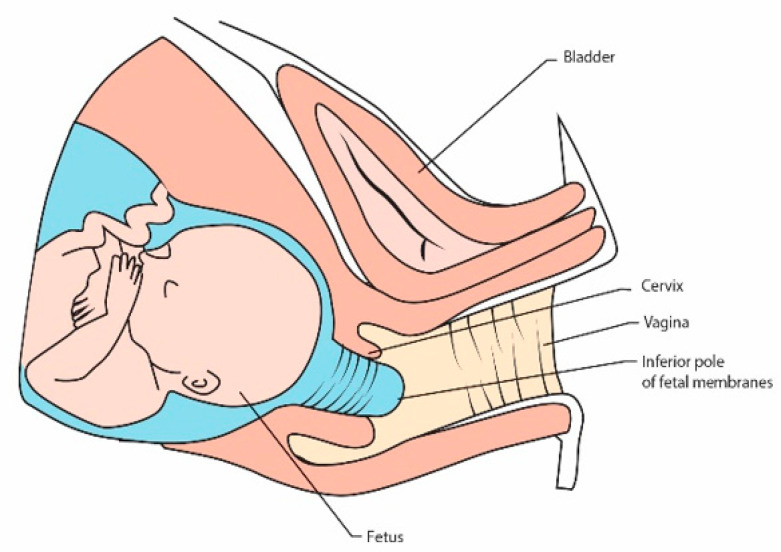
Cervical insufficiency with protruding membranes into vagina.

**Figure 2 jcm-10-01270-f002:**
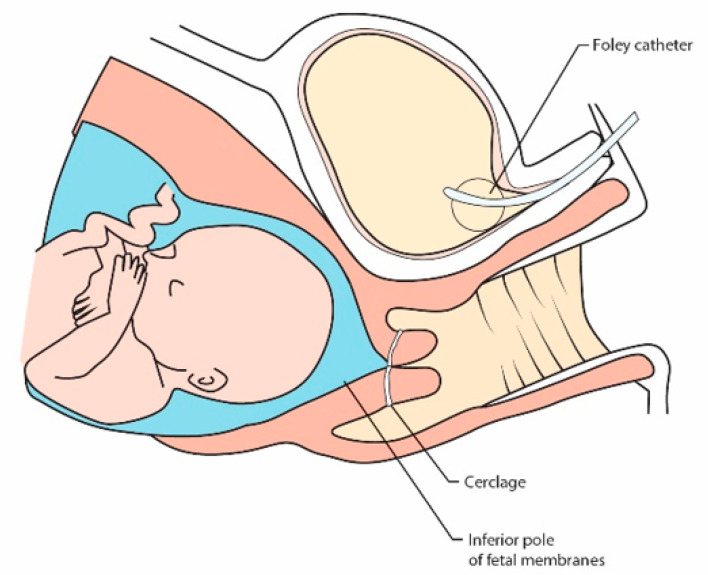
Filling the bladder with physiological saline in a patient positioned in the Trendelenburg position turned out to be technically effective for draining a prolapsed fetal membranes.

**Figure 3 jcm-10-01270-f003:**
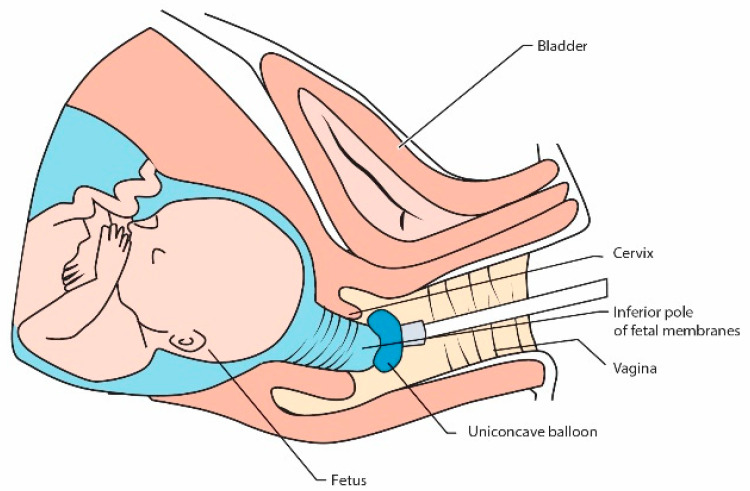
Using a uniconcave balloon device for repositioning fetal membranes into the uterus during emergency cerclage [63].

**Table 1 jcm-10-01270-t001:** Emergency suture effectiveness.

		Prolongation of Pregnancy	Gestational Age at Delivery	Delivery Before 32 Weeks 34 Weeks	Mean Birth Weight	Neonatal Survival
Althuisius et al. [72]	Cerclage group (n = 13)	54 days	29.9 weeks	53.8%	N/A	56.2%
	Bed rest group (n = 10)	20 days *	25.9 weeks NS	100% *		28.6% NS
Aoki et al. [73]	Cerclage group (n = 15)	44 days	32.4 weeks	33.3%	N/A	N/A
	Bed rest group (n = 20)	12.5 days **	26.0 weeks *	90% **		
Daskalakis et al. [74]	Cerclage group (n = 29)	8.8 weeks	N/A	31%	2101 g	96%
	Bed rest group (n = 17)	3.1 weeks ***		94.1% ***	739 g ***	57.1% *
Stupin et al. [50]	Cerclage group (n = 89)	41 days	28.0 weeks	N/A	1340 g	72%
	Bed rest group (n = 72)	3 days ***	23.0 weeks ***		750 g ***	25% ***
Bayrak et al. [75]	Cerclage group (n = 27)	64 days	31.5 weeks	51.9%	N/A	63%
	Bed rest group (n = 8)	13.5 days **	24.1 weeks **	100% **		0% **
Ciavattini et al. [76]	Cerclage group (n = 18)	16.8 weeks	34.8 weeks	16.7%	2814 g	100%
	Bed rest group (n = 19)	7.2 weeks ***	26.7 weeks **	21.1% NS	1482 g ***	100% ^#^
Costa et al. [77]	Cerclage group (n = 19)	48.6 days	28.6 weeks	N/A	1468.3 g	47.4%
					861.2 g NS	36.3% NS
	Bed rest group (n = 11)	16 days *	23.3 weeks *			

^#^ determined on the total number of live births; * *p* < 0.05; ** *p* < 0.01; *** *p* < 0.00; N/A not available; NS—not statistically significant.

**Table 2 jcm-10-01270-t002:** Risk factors of the emergency suture failure.

	Risk Factor
1	Primigravidas
2	Multigravidas with a history of second-trimester pregnancy loss
3	Cervical dilatation ≥ 4 cm
4	Bulging membranes into the vagina
5	Infection
6	The presence of myeloperoxidase, lactoferrin, glucose-6-phosphate isomerase, lipocalin-2, and lymphocyte cytosolic protein 1 in amniotic fluid
7	Multiple gestation
8	Level of fetal fibronectin over 500 ng/mL

**Table 3 jcm-10-01270-t003:** Factors associated with early preterm delivery in the multiple logistic regression model and number of points contributed to the score by each factor. Reprinted with permission from ref. [64]. Copyright 2012 Wiley.

Variable Adjusted 95% Confidence Score Odds Ratio Interval Points			
**Obstetric History**			
**Multigravidas without history of STPL**	**1**		0
Primigravidas	4.8	1.1–23.6	4
Multigravidas with history of STPL	7.5	1.3–43.9	5
**Cervical Dilatation**			
1 cm	1		0
2 cm	1.4	1.1–2.3	1
3 cm	2.0	1.2–5.5	2
≥4 cm	4.1	1.9–30.0	4
**Membranes**			
Visible at external os	1		0
Bulging into the vagina	4.2	1.1–16.8	4
**Infection WBC ≥ 13,600/ mm^3^ or C-Reactive Protein > 15 mg/L**			
No	1		0
Yes	2.3	1.5–7.8	2

STPL-second-trimester pregnancy loss; WBC–white blood cells.

## Data Availability

MDPI Research Data Policies.

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
