# Peer review of "Emergency Cervical Cerclage"

_jcm, 2021, doi:10.3390/jcm10061270_

Round 1
Reviewer 1 Report
The authors have made considerable revisions to the initial submission. I have no additional comments.
Author Response
We would like to express our gratitude to the Reviewer for all previous and current helpful comments which have made a substantial contribution to the quality of our paper.
Thank you very much for your comment.
Reviewer 2 Report
The figures are of a good standard and the tables deliver the necessary information in a manageable format.
On line 221 it mentions 3600 studies were identified and 38 analysed.
I assume that this should probably be 3600 study participants (patients) rather than actual studies
Author Response
We would like to express our gratitude to the Reviewer for all previous and current helpful comments which have made a substantial contribution to the quality of our paper.
We apologize for the confusion caused. Chatzakis et al. [39] informed in the manuscript part entitled “Data search results” that: “The electronic search from the databases yielded 3607 potential studies, of which 3564 were excluded because they were duplicates, or a review of the title or abstract that did not meet the inclusion criteria, leaving 38 studies for full-text review. After the full manuscript review, we finally considered 13 studies […]. One study was not included in the quantitative analysis because it reported only one outcome of interest and the control group consisted only of two women. All the included studies were nonrandomized trials. The excluded studies, along with the reason for their exclusion are shown in the Supporting Information.”
Regarding your comment, we have revised our previous sentences:
“A meta-analysis by Christos Chatzakis and co-authors was published in 2020. The authors analyzed 3607 studies regarding emergency cervical cerclage, reviewed 38 of them and assumed 12 observational analysis with 1021 patients [39].”
as follows:
“A meta-analysis by Christos Chatzakis and co-authors was published in 2020. The authors reviewed 38 studies regarding emergency cervical cerclage and assumed 12 observational analysis with 1021 patients [39].”
Reviewer 3 Report
I thank the Editor for the opportunity to revise the review entitled “ Emerging cervical cerclage". The manuscript is interesting, the subject is important in the field showing interesting results about the role of cervical cerclage.
In this paper the authors review the role of emerging cervical cerclage. Minor comments:
Please, possibly the paper would be improved after the addition of a section about the role of infective agents in the context of cervical insufficience and the possible role of antibiotic prophylaxis associated to the procedure of cervical cerclage.
Author Response
We would like to express our gratitude to the Reviewer for all previous and current helpful comments which have made a substantial contribution to the quality of our paper.
Thank you for your meaningful comments. Infection represents one of main risk factors of the emergency suture failure. We presented this problem in the part entitled “4. Risk Factors of the Emergency Suture Failure”:
“Appropriate perioperative procedures may reduce the risk of spontaneous premature delivery. Unfortunately not all risk factors can be eliminated Table 2, there are beyond the possibilities of the operators. It seems that the only modifiable factor is treatment of genital tract infections. Undoubtedly, the most successful situation is a lack of vaginal colonization with pathogenic microorganisms and the absence of signs of infection. Considering that operators are obligated to make a quick decision about qualifying patient to rescue cerclage taking into account the increasing dilatation, targeted treatment in the preoperative period is usually impossible.
Fuchs et al. used multivariate logistic regression methods to develop a score for assessing the risk of early preterm delivery before 32 weeks in women with singleton pregnancies receiving emergency cervical cerclage [50]. The score, ranging from 0 to 15 points, was based on the following four criteria independently associated with early preterm delivery: obstetric history; cervical dilatation; membranes bulging into the vagina; and infection Table 3. Each score value was associated with a predicted probability of early preterm birth. The authors found that a history of second-trimester pregnancy loss, nulliparity, a cervix dilated more than 4 cm, membranes bulging into the vagina, and infection (i.e., white blood cells (WBC) ≥ 13 600/ mm3 or C-reactive protein (CRP) > 15 mg/L) are associated with emergency suture failure [50].
Ito et al. demonstrated that elevated serum inflammatory markers before the procedure are associated with its failure [66]. CRP value and WBC are recognized predictors of subclinical chorioamnionitis [67]. The authors observed that peripheral CRP levels (≥ 4 mg/l) and WBC counts (≥ 10,000/mm3) were associated with a significantly decreased likelihood of delivery at and after 28 weeks gestation [66].
Some authors expect that Gram’s method and culture of amniotic fluid are imperfect techniques to detect infection in patients with cervical insufficiency that are qualified to emergency cerclage procedure [68]. Proteomic profiling of amniotic fluid could be better solution in that cases. There are different expression of proteins between patients with delivery before one week from inserting cerclage and those which had delivery later. Patients with shorter prolongation of pregnancy had activated inflammatory response, chemotaxis of immune cells, and inhibited bacterial growth. These preliminary results indicate that the proteomic profiling of amniotic fluid may be an effective predictor of cervical insufficiency outcome [68]. More research on this method is required.”
Regarding the Reviewers’ suggestion we have added sentences and references as follows:
“Previous studies have found that microorganisms are present in the amniotic cavity in 8-52% of patients with cervical insufficiency [41-48]. Oh et al. [49] concluded that administration of antibiotics (ceftriaxone, clarithromycin, and metronidazole) in patients with cervical insufficiency and intra-amniotic inflammation or intra-amniotic infection can improve the treatment success.
On the other hand, Chatzakis et al. in their meta-regression analysis showed that there was no effect of antibiotics administration on the outcome of pregnancy prolongation [39]. Most studies reported that antibiotics administration was at the discretion of the managing clinicians [50-54]. Due to lack of randomized control trials, these observations should be viewed with caution, as the quality of evidence is low to very low.”
References:
41. Goodlin, R.C. Cervical incompetence, hourglass membranes, and amniocentesis. Obstet. Gynecol. 1979, 54, 748-750.
42. Romero, R.; Gonzalez, R.; Sepulveda, W.; Brandt, F.; Ramirez, M.; Sorokin, Y; Mazor, M.; Treadwell, M.C.; Cotton, D.B. Infection and labor. VIII. Microbial invasion of the amniotic cavity in patients with suspected cervical incompetence: prevalence and clinical significance. Am. J. Obstet. Gynecol. 1992, 167, 1086-1091.
43. Mays, J.K.; Figueroa, R.; Shah, J.; Khakoo, H.; Kaminsky, S.; Tejani, N. Amniocentesis for selection before rescue cerclage. Obstet. Gynecol. 2000, 95, 652-655.
44. Lee, S.E.; Romero, R.; Park, C.W.; Jun, J.K.; Yoon, B.H. The frequency and significance of intraamniotic inflammation in patients with cervical insufficiency. Am. J. Obstet. Gynecol. 200, 198, 633.e1-633.e8.
45. Bujold, E.; Morency, A.M.; Rallu, F.; Ferland, S.; Tétu, A.; Duperron, L.; Audibert, F.; Laferrière, C. Bacteriology of amniotic fluid in women with suspected cervical insufficiency. J. Obstet. Gynaecol. Can. 2008, 30, 882-887.
46. Airoldi, J.; Pereira, L.; Cotter, A.; Gomez, R.; Berghella, V.; Prasertcharoensuk, W.; Rasanen, J.; Chaithongwongwatthana, S.; Mittal, S.; Kearney, E.; Tolosa, J.E. Amniocentesis prior to physical exam-indicated cerclage in women with midtrimester cervical dilation: results from the expectant management compared to Physical Exam-indicated Cerclage international cohort study. Am. J. Perinatol. 2009, 26, 63-68.
47. Oh, K.J.; Lee, S.E.; Jung, H.; Kim, G.; Romero, R.; Yoon, B.H. Detection of ureaplasmas by the polymerase chain reaction in the amniotic fluid of patients with cervical insufficiency. J. Perinat. Med. 2010, 38, 261-268.
48. Lisonkova, S.; Sabr, Y.; Joseph, K.S. Diagnosis of subclinical amniotic fluid infection prior to rescue cerclage using gram stain and glucose tests: an individual patient meta-analysis. J. Obstet. Gynaecol. Can. 2014, 36, 116-122.
49. Oh, K.J.; Romero, R.; Park, J.Y.; Lee, J.; Conde-Agudelo, A.; Hong, J.S.; Yoon, BH. Evidence that antibiotic administration is effective in the treatment of a subset of patients with intra-amniotic infection/inflammation presenting with cervical insufficiency. Am. J. Obstet. Gynecol. 2019, 221, 140.e1-140.e18.
50. Stupin, J.H.; David, M.; Siedentopf, J.P.; Dudenhausen, J.W. Emergency cerclage versus bed rest for amniotic sac prolapse before 27 gestational weeks. A retrospective, comparative study of 161 women. Eur. J. Obstet. Gynecol. Reprod. Biol. 2008, 139, 32–37.
51. Pereira, L.; Cotter, A.; Gómez, R.; Berghella, V.; Prasertcharoensuk, W.; Rasanen, J.; Chaithongwongwatthana, S.; Mittal, S.; Daly, S.; Airoldi, J.; Tolosa, J.E. Expectant management compared with physical examination-indicated cerclage (EM-PEC) in selected women with a dilated cervix at 14(0/7)-25(6/7) weeks: results from the EM-PEC international cohort study. Am. J. Obstet. Gynecol. 2007, 197, 483.e1-483.e8.
52. Ko, H.S.; Jo, Y.S.; Kil, K.C.; Chang, H.K.; Park, Y.G.; Park, I.Y.; Lee, G.; Kim, S.; Shin, J.C. The clinical significance of digital examination-indicated cerclage in women with a dilated cervix at 14 0/7-29 6/7 weeks. Int. J. Med. Sci. 2011, 8, 529-536.
53. Gimovsky, A.C.; Suhag, A.; Roman, A.; Rochelson, B.L.; Berghella, V. Pessary versus cerclage versus expectant management for cervical dilation with visible membranes in the second trimester. J. Matern. Fetal Neonatal Med. 2016, 29, 1363-1366.
54. Ventolini, G.; Genrich, T.J.; Roth, J.; Neiger, R. Pregnancy outcome after placement of 'rescue' Shirodkar cerclage. J. Perinatol. 2009, 29, 276-279.
This manuscript is a resubmission of an earlier submission. The following is a list of the peer review reports and author responses from that submission.
Round 1
Reviewer 1 Report
The article by Wierzchowska-Opoka reviews the clinical value of emergency cervical cerclage. The matter is important. The overall structure of the article is fair. There are several points that should be improved. In particular, taking into account that the article sould be included in a special issue on recurrent pregnancy loss, it would be important discussing the isssues concerning:
- (emergency) cervical cerclage with specific application to RPL.
- The relevance of cervical insufficiency in women with RPL
There are other points, mainly concerning language.
Line 27: remove “phenomena of”
Line 45: “Puberty” ??
Lines 51-52: check and improve language
Line 62: remove “pregnancy complicated with”
Line 89: I believe that the word “egg” is improper
Line 90 : ”moving the fetal bladder away” ??
In the section “Emergency suture procedure” more detailed information should be given on the technique(s) by which the sutures are applied, also with the help of an additional figure. This because the article should be published in a journal that is not only aimed at ob/gyn readres, but at a more extensive group of readers, which cannot be confident with surgical procedures.
Fig. 1: I suggest to clearly indicate the inferior pole of the fetal membranes
Use the word “regimen” instead of “regime”
Line 203: I suggest to remove the words “very interesting”. Let judge whether it is interesting to the reader
The considerations about causes and determining factors of cervical insufficiency should be expanded.
Line 217: please check the language
Lines 228-236: Adding a table showing the quoted score by Fuchs and coll. would improve the quality of the information provided (check whether permission is required)
Line 264: remove “cited”
Lines 270-1: please check the language
Line 273: remove “laden”
Reviewer 2 Report
The authors attempted to summarize the state-of-the-art treatment against cervical insufficiency--which the authors note is a significant contributor to pregnancy loss. The authors review the procedure of cervical cerclage which is an effective treatment against pregnancy loss associated with cervical insufficiency, variations and precursors of the procedure itself, and possible risk factors.
Comments
- The article suffers from grammatical errors throughout that make it very difficult to read, a major revision to the grammar is needed prior to acceptance for publication
- The novelty and timeliness of the article needs to be carefully evaluated, in 2017 Alfirevic et al. published a systematic review of the literature on cervical cerclage for the prevention of preterm birth. A comparative review of these two articles make it difficult to justify this publication given the lack of new insights provided by the article, in addition to the lack of clear meta-analysis and methodology for the inclusion/exclusion of studies. If the authors can clearly justify how this review advances the field over existing systemic reviews, please clearly state this during the introduction to identify the gap of knowledge in the field
- The illustration of cervical cerclage needs to be digitized and not drawn by hand, labels of key components should be clearly positioned (the fetus, the cervix especially should be labeled right over the appropriate component of the diagram)
Reviewer 3 Report
The topic is relevant and this review presents evidence that identifies emergency / rescue cerclage as a successful procedure in carefully selected individuals. It highlights the lack of randomised control trials in this area and the need for further definitive studies. Such trials are challenging to complete and current efforts such as C Stitch 2 are proving difficult.
There are a number of areas throughout the manuscript where the authors intended meaning is not reflected in the English language used and these need to be re-written. I have highlighted these beneath.
Introduction line 29-30: development in neonatal care has led to a spectacular increase in neonatal survival, and has also significantly reduced their incidence
I think the second half of this sentence which I have highlighted can be deleted
Line 30-31 sentence needs re-arranging
Line 38 this complications should read these complications
Line 38-39 the inability of the apparatus closing the uterus to maintain pregnancy does not read well
Line 42 Fetal egg is not a recognised term and needs to be replaced throughout the entire manuscript
Line 45 Puberty is not the correct word here
Line 52 does not read well
Line 54 abrasion of the uterine cavity is not a recognised term and I dont understand the phrase improperly stocked postpartum cervical rupture
Line 57 sentence needs re-arranging
Line 64, 65,66 does not read well
Line 70 does not read well
Line 81 does not read well
Line 89 "egg" needs replacing
Line 90 "gently moving the fetal bladder"- this phrase does not make sense
Line 109 "useful toll" confusing term
Line 127 "bed regime" I assume this is bed rest but not clear and needs to be replaced or expanded upon as it is used multiple times in the paper
Line 155-158. Lines 155+ 156 are repeated in 157 + 158
Line 164 does not read well
Line 239 Reference required for the statement CRP and WCC are recognised predictors or subclinical chorioamnionitis. Although used clinically I am not aware of any studies identifying a WCC or CRP cut-off for chorioamnionitis
Line 253 "verify" verified
Line 258 "large dilatation" does not read well
Line 265 does not read well
Line 273 "laden" can be removed
Line 284 does not read well
Line 287-288 does not read well
The figure presented in section 2. Emergency suture procedure is not of publication standard.
One method extensively used is to replace the fetal membranes with a partially inflated foley catheter which is then completely inflated to 20-30ml at the internal cervical os. The stitch is then placed around the cervix circumferentially and pulled tight as the foley catheter is simultaneously deflated and removed.
Overall in its current format I do not think that this manuscript is of publication standard. It would need to be re-written with particular attention given to the lines highlighted above with more in-depth analysis and critique of the individual studies presented.
Round 2
Reviewer 1 Report
The authors made major revisions resulting in an improvement of the article.
Reviewer 2 Report
The authors have addressed many major concerns with regards to the novelty of the study, and have also made quality of life changes to the text and figures. There continues to be minor grammatical issues that should be addressed.
Reviewer 3 Report
I recognise that considerable improvements have been made from the first submission. There are still some English language issues detailed below and the section describing the effectiveness of rescue cerclage seems more like a list of studies and their findings rather than an in-depth analysis of their findings.
Line
12- change this - these
13- change Manifesting in - manifest as
20 - change newborns - newborn
change- However, it doesn’t increase the risk of chorioamnionitis and preterm premature rupture of membranes. - without increasing the risk of chorioamnionitis and PPROM
32 - change increased survival of very preterm babies
41- change abortion - miscarriage
49- change Prevent fetal membranes from falling out of the uterus. Thanks to the mucus it is a barrier cervix protects against ascending infections
- Prevents exposure and prolapse of the fetal membranes and in combination with the mucus plug protects against ascending infection
53 - change dilating - dilatation of the cervix
54 - change - during the labor, but when initiated before the 37th week of pregnancy -
during labour but when initiated before 37 weeks of pregnancy
55 - change-It depends on several factors -
Several factors are implicated in cervical insufficiency including;
70- EMCS at full dilatation is also implicated in subsequent cervical insufficiency / PTB / Second trimester miscarriage
76- change whites - white women
79- change blacks - black women
81- change protection - Protecting
103 - A dilated cervix is identified on speculum examination / physical examination or transvaginal ultrasound
108- Sentence needs re-wording
122 - the aim is to
Figure 1 - much improved
148 - change- but the results so far do not provide spectacular achievements - but the results are disappointing
160- change- is a useful device
161- change-effectively replaced fetal membranes into the uterine cavity to enable the placement of an emergency cerclage
169- References 43 and 52 are contradictory needs further explaination
179- Change the best rest – bed rest
203- change – before the 32 week – before 32 weeks
210- change – 22 patients were qualified – 22 patients qualified
213- delivery occurred at 28 weeks on average
222 – change -Protruding – Protrusion into the vagina
226– change –terminated the pregnancy- Delivered
239- 25th week of pregnancy
242- Rescue suture has been recommended by
249 – Pregnancy prolongation
253- delete- and increase the chance of its successful solution
254- delete- A very interesting meta-analysis was published in 2020. Add- A meta-analysis by
255- delete – fully
268 – delete - Unfortunately some risk factors for the failure effectiveness of the
270- delete- of the emergency suture failure
273 – not sure this is the right word – Awkwardly
317- change – level - levels
